# Molecular Detection of Drug-Resistance Genes of *bla*_OXA-23_*-bla*_OXA-51_ and *mcr-1* in Clinical Isolates of *Pseudomonas aeruginosa*

**DOI:** 10.3390/microorganisms9040786

**Published:** 2021-04-09

**Authors:** Fabiana Nitz, Bruna Oliveira de Melo, Luís Cláudio Nascimento da Silva, Andrea de Souza Monteiro, Sirlei Garcia Marques, Valério Monteiro-Neto, Rosimary de Jesus Gomes Turri, Antonio Dantas Silva Junior, Patrícia Cristina Ribeiro Conceição, Hilário José Cardoso Magalhães, Adrielle Zagmignan, Thiago Azevedo Feitosa Ferro, Maria Rosa Quaresma Bomfim

**Affiliations:** 1Laboratório de Biologia Molecular de Microrganismos Patogênicos, Universidade Ceuma, São Luís 65075-120, Brazil; fabi_nitz@hotmail.com (F.N.); brunaoliv96@gmail.com (B.O.d.M.); valerio.monteiro@ufma.br (V.M.-N.); adantasjr@uol.com.br (A.D.S.J.); hicard@hotmail.com (H.J.C.M.); thafeitosaf@hotmail.com (T.A.F.F.); 2Laboratório Cedro, São Luís 65020-570, Brazil; sirleigmarques@gmail.com (S.G.M.); patriciacsribeiro@hotmail.com (P.C.R.C.); 3Laboratório de Patogenicidade Microbiana, Universidade Ceuma, São Luís 65075-120, Brazil; adrielle.zagmignan@ceuma.br; 4Laboratório de Microbiologia Aplicada, Universidade Ceuma, São Luís 65075-120, Brazil; andreasmont@gmail.com; 5Programa de Pós-Graduação em Ciências da Saúde, Universidade Federal do Maranhão, São Luís 65085-582, Brazil; 6Departamento de Farmácia, Centro de Ciências Biológicas e da Saúde, Universidade Federal do Maranhão, São Luís 65080-805, Brazil; rositurri@gmail.com

**Keywords:** extensive drug resistance (XDR), oxacillinase genes, *Pseudomonas aeruginosa*, *mcr-1* gene

## Abstract

*Pseudomonas aeruginosa* has caused high rates of mortality due to the appearance of strains with multidrug resistance (MDR) profiles. This study aimed to characterize the molecular profile of virulence and resistance genes in 99 isolates of *P. aeruginosa* recovered from different clinical specimens. The isolates were identified by the automated method Vitek2, and the antibiotic susceptibility profile was determined using different classes of antimicrobials. The genomic DNA was extracted and amplified by multiplex polymerase chain reaction (mPCR) to detect different virulence and antimicrobial resistance genes. Molecular typing was performed using the enterobacterial repetitive intergenic consensus (ERIC-PCR) technique to determine the clonal relationship among *P. aeruginosa* isolates. The drug susceptibility profiles of *P. aeruginosa* for all strains showed high levels of drug resistance, particularly, 27 (27.3%) isolates that exhibited extensively drug-resistant (XDR) profiles, and the other isolates showed MDR profiles. We detected the polymyxin E (*mcr-1*) gene in one strain that showed resistance against colistin. The genes that confer resistance to oxacillin (*bla*_OXA-23_ and *bla*_OXA-51_) were present in three isolates. One of these isolates carried both genes. As far as we know from the literature, this is the first report of the presence of *bla*_OXA-23_ and *bla*_OXA-51_ genes in *P. aeruginosa*.

## 1. Introduction

Among opportunistic microorganisms that cause human infections, the bacterium *Pseudomonas aeruginosa* remains one of the most prevalent agents involved in outbreaks of infections in the hospital environment worldwide [1].This pathogen is the main species from the genus *Pseudomonas*, which comprises 144 species [2], where approximately 25% are related to human infections [3]. *P. aeruginosa* infects several anatomic sites such as the urinary, respiratory, and gastrointestinal tracts; skin; soft tissues; blood; bones; and eyes and causes a variety of systemic infections, especially in patients with severe burns and immunosuppression [3,4].

The pathogenic arsenal of *P. aeruginosa* comprises a range of virulence factors responsible for the neutralization of host defenses, induction of tissue damage, and biofilm formation and increases the competitiveness of this microorganism [5,6]. Other important virulence factors in *P. aeruginosa* are flagella, fimbriae, superficial polysaccharides, and pili-type IV that are involved in bacterial colonization [7]. In addition, *P. aeruginosa* is naturally resistant to several types of antimicrobials commonly used in clinical practice due to the presence of a lipopolysaccharide-rich outer membrane that surrounds the bacterial cell and prevents the entry of certain antimicrobials [8,9]. Other factors also contribute to this panorama, such as those related to low permeability of the outer membrane, the efflux system (that actively expels the antibiotics out of the cell), and the production of antimicrobial-inactivating enzymes [5,6,10,11,12].

Based on the resistance profile, the bacterial strains could be classified as multidrug-resistant (MDR) and extensively drug-resistant (XDR) strains. The MDR profile is defined as resistance to at least one drug in three or more categories of antimicrobials tested, while XDR is defined as non-susceptibility to at least one agent in all but two or fewer antimicrobial categories (i.e., bacterial isolates remain susceptible to only one or two categories). Pan-drug resistance is defined as non-susceptibility to all agents in all antimicrobial categories [13].

Antimicrobial resistance is responsible for serious clinical consequences and significant socioeconomic losses related to increased morbidity and mortality of patients [14]. Epidemiological studies worldwide have revealed that *P. aeruginosa* are among the microorganisms most frequently involved in bloodstream infections (58.5%)[15]. Moreover, this bacterium is responsible for 10% of all nosocomial-acquired infections [16], with a mortality rate of approximately 70.0% [17].

Due to the clinical importance of this pathogen, the detection of virulence genes and resistance towards antimicrobials most used in clinical practice is of great interest in terms of public health and may contribute to the control of dissemination of multi-resistant isolates in the hospital environment. Herein, the phenotypic and molecular characterization of antimicrobial resistance and the occurrence of virulence factor are detected in 99 *P. aeruginosa* strains recovered from clinical samples obtained from different hospitals located in São Luís (Maranhão, Brazil).

## 2. Methods

### 2.1. Isolation of Pseudomonas aeruginosa 

A total of 99 of clinical isolates were provided by routine service of two private laboratories of clinical analysis of São Luís-MA. These isolates were recovered from clinical specimens collected from November 2015 to August 2016 from patients attending different hospitals in the city of São Luís, Maranhão, Brazil. The epidemiological data (gender and anatomical site of origin) were provided by the hospitals. Initially, these samples were processed in the clinical microbiology section of the two laboratories, where the bacterial strains were isolated with Mueller–Hinton agars (MH, Difco, Laboratories, Detroit, MI, USA) and brain heart infusion broth (BHI; Difco, Laboratories, Detroit, MI). 

### 2.2. Antimicrobial Susceptibility by the Automated Method

The identification of bacterial isolates and their susceptibility profiles were determined by the Vitek 2 Compact system (bioMérieux, Marcy l’Etoile, France) using, respectively, a Gram-negative (GN) card for bacterial identification (GN ID) and antimicrobial susceptibility testing (AST-N239 cards), according to the instructions of the manufacturer. All isolates were identified as being *Pseudomonas aeruginosa,* and aliquots of each were stored in BHI plus 15% glycerol kept in the freezer at −80 °C for later tests. Regarding the clinical specimens from which these bacterial isolates were recovered, 40 were isolated from bloodstream infections, 37 were from tracheal secretions, 11 were from wound fragments, and 11 were from urine samples. 

### 2.3. Antimicrobial Susceptibility Testing by Disk Diffusion

In the Laboratory of Molecular Biology of Microorganisms of the Ceuma University, the bacterial strains were checked for their purity and biochemical tests were performed to confirm the identification of all bacterial isolates. In addition, to confirm the susceptibility pattern obtained in the automated method, the Mueller–Hinton agar diffusion test (Kirby–Bauer) was performed with discs (Oxoid^®^ Limited, Basingstoke, UK) for the following antimicrobial classes: ß-Lactamics Ampicillin (AMP; 10 μg), Ampicillin/Sulbactam (ASB; 10 μg/10 μg), Aztreonam (AZT; 30 μg), Cefepime (CEF, 30 μg), Cefotaxime (CTX; 30 μg), Cefoxitin (CFO; 30 μg), Ceftazidime (CAZ; 30 μg), and Piperacillin/Tazobactam (PTZ; 100 μg/10 μg); Aminoglycosides, Amikacin (AMI; 30 μg), and Gentamicin (GEN; 10 μg); Quinolones, Ciprofloxacin (CIP; 5 μg); Glycylcyclines (TIG; 15 μg); Carbapenems, Imipenem (IMP; 10 μg), and Meropenem (MER; 10 μg); Polymixin B (POLII; 300 U) and Polymixin E, Colistin (POLI; 10 μg). The diameters of the zones of inhibition were measured, and the results were interpreted in accordance with the criteria of the Clinical and Laboratory Standards Institute [18].

### 2.4. Total DNA Extraction

Total genomic DNA from all clinical isolates was obtained using a MagaZorb^®^ DNA Mini-Prep Kit (Promega, a Corporation, Madison, WI-USA) according to the manufacturer’s instructions. The concentration and purity of the extracted DNA were checked using a Nanodrop-ND1000 (Thermo Fisher Scientific, Waltham, MA, USA). The DNA from the standard strain *P. aeruginosa* ATCC 27,853 was also extracted to be used as a positive control.

### 2.5. Molecular Detection of Genes Related to Virulence and Antimicrobial Resistance

Multiplex PCR (mPCR) and specific PCR assays were employed for the detection of genes that codify virulence factors and proteins associated with antimicrobial resistance. The sequence of each primer and the distribution into the groups for amplification by mPCR are shown in Table 1 and Table 2. For detection of the *mcr-1* gene, we designed a pair of primers that amplify a fragment of 400 base pairs. All PCR reactions were performed on a Mycycler Bio Rad model 580BR3578 thermocycler, with a final volume of 25 μL that consisted of 1 μL of each specific primer (10 pmol/L) plus 12.5 μL Master Mix PCR (Promega^®^) [Taq DNA polymerase (dNTPs, MgCl_2_, PCR buffer (pH 8.5))], 1.5 μL of nuclease-free water, and 3 μL of the bacterial lysate containing the DNA. The amplification reactions were executed under the following conditions: 94 °C for 5 min (initial denaturation) followed by 30 cycles of 94 °C for 1 min; annealing for 1 min at temperatures ranging from 49 °C to 66 °C (depending on primer set used); extension at 72 °C for 1 min; and final extension at 72 °C for 7 min.

The amplification efficiency was determined by electrophoresis applying 8 μL of each PCR product on agarose gels (1.5% (*w*/*v*) in Tris-acetate-EDTA buffer (TAE: 40 mM Tris-acetate and 1 mM EDTA)). A 100 bp DNA ladder (Promega Corporation, Madison, USA) was included in each run. After electrophoresis, the agarose gels were stained with ethidium bromide (0.5 μg/mL) and photographed under ultraviolet light (UV) at 260 nm.

### 2.6. Sequencing of PCR Products

To confirm the presence of important antimicrobial resistance genes (*bla*_OXA-51_, *bla*_OXA-23_, and *mcr-1*) in isolates positive for these genes, the amplified products were purified using the commercial kits: Wizard^®^ SV Gel and PCR Clean-Up System Kit (Promega Corporation, Madison, USA). In addition, to prove that these three isolates were indeed *Pseudomonas aeruginosa*, we amplified and sequenced the 16S region of these isolates using bacterial Universal primers: 27F (5′-AGAGTTTGATCATGGCTCAG-3′) and 1492R (5′-GGTTACCTTGTTACGACTT-3′) following the protocol previously described [29]. The sequencing of purified PCR products was performed with ABI Prism BigDye Kit on the ABI 3130 Genetic Analyzer (Applied Biosystems). The samples were sequenced at least three times in each direction of the tape, making a total of six sequences from the same sample. All sequencing assays were done by Myleus Biotecnologia, based in Belo Horizonte, MG. The sequence electropherograms were analyzed with ChromasPro software (http://www.technelysium.com.au/chromas.html (accessed on 23 May 2020)). The similarity between the sequences was verified with BLASTn tool (Basic Alignment Search Tool–(http://www.ncbi.nlm.nih.gov/BLAST/ (accessed on 23 May 2020).

### 2.7. Genotyping Using Enterobacterial Repetitive Intergenic Consensus (ERIC)-PCR

The clonal diversity of the *P. aeruginosa* isolates was assessed by Enterobacterial Repetitive Intergenic Consensus (ERIC)-PCR using the primer proposed by Versalovic et al. al. [30]. The reactions were performed with 100 ng of genomic bacterial DNA, 10 pmol of the primers ERIC-1R (5′-ATGTAAGCTCCTGGGGATTCAC-3′) and ERIC-2 (5′-AAGTAAGTGACTGGGGTGAGCG-3′) (Invitrogen, Carlsbad, CA, USA), plus 12.5 μL of Master Mix (Promega Corporation, Madison, USA) and ultra-pure water to bring the final reaction volume to 25 μL. The ERIC-PCR reactions were performed under the following conditions: initial denaturation at 94 °C for 3 min, followed by 35 cycles of denaturation (at 94 °C for 30 s), annealing (52 °C for 1 min), and extension (72 °C for 2 min), with a final extension at 72 °C for 6 min. PCR amplicons were submitted to electrophoresis using a 1.5% (*w*/*v*) agarose gel in acetate-EDTA buffer (TAE: 40 mM Tris-acetate and 1 mM EDTA) at 85 V for 1 h and 30 min. A 100 bp DNA ladder (Promega Corporation, Madison, WI, USA) was included in each run. After electrophoresis, the gels were stained with ethidium bromide (0.5 μg/mL) and photographed under UV at 260 nm.

The ERIC-PCR profiles were scored by fragment size with the LabImage-1D gel analysis software, Version 3.2 (1D: V6.2. Available online: http://www.kapelanbio.com/ (accessed on 23 May 2020)). Amplified fragments were scored as absent (0) or present (1) to construct a dendrogram. ERIC-PCR genotype dendrograms were constructed based on the average similarity of the matrix using the unweighted pair group method with the arithmetic mean and the Sørensen–Dice similarity coefficient using NTSYS-pc version 2.1, Exeter Software (New York, NY, USA). The nearest neighbor-joining clustering method was used to show relationships between similar groups.

### 2.8. Statistical Analysis of the Data

The data were analyzed by the ANOVA test using the statistical program IBM SPSS Statistics 20. Initially, descriptive statistics were made, i.e., frequency table of variables (genes) were analyzed. The significance level (α) was 5%, i.e., it was considered significant when *p* < 0.05. The contingency coefficient C test by the Biostat 5.0 program with a 95% confidence interval and a *p* < 0.05 value and the Kruskal Wallis test were also used for statistical analysis.

### 2.9. Ethical Aspects

The study was approved by the Research Ethics Committee of Universidade CEUMA in compliance with the requirements of Resolution 466/2012 of the Brazil National Health Council, under Consubstantiated opinion number 3.540.095.

## 3. Results

### 3.1. Characterization of Samples and Drug Susceptibility Profile

In the present study, 99 *P. aeruginosa* isolates were recovered from the following clinical specimens: wound tissue fragment (*n* = 11/11.1%), urine (*n* = 11/11.1%), tracheal secretion (*n* = 37/37.4%), and blood (*n* = 40/40.4%). Most *P. aeruginosa* strains (*n* = 51/51.5%) were obtained from the male gender, and regarding age for both groups, the strains were more prevalent (*n* = 62/62.6%) in patients above 60 years of age (in this group, 51.5% were men), considering the contingency coefficient C; *p* < 0.05 (Table 3).

The drug susceptibility profiles of the tested strains determined by the Vitek 2 Compact system are shown in Table 4. We observed that all *P. aeruginosa* isolates showed resistance towards ampicillin/sulbactam, cefoxitin, and tigecycline while all isolates were sensitive to polymyxin B, but one strain showed resistance against colistin (polymyxin E). A high prevalence of strains had resistance towards β-lactam antibiotics piperacillin/tazobactam (74.7%) and cephalosporins, with seventy isolates (70.7%) resistant to ceftazidime (3rd generation cephalosporins) and 61 strains (61.6%) resistant to cefepime (4th generation cephalosporins). It is important to highlight that 53.5% isolates were resistant to both cephalosporins. Most of the strains also exhibited resistance against ciprofloxacin (66.7%), gentamicin (58.6%), carbapenems imipenem (57.6%), and meropenem (56.6%).

In general, all strains showed high levels of drug resistance, particularly, 27 (27.3%) isolates showed XDR profiles. Of this total, 26 were resistant to six different classes of antimicrobials and one isolate was resistant to all seven classes of antibiotics. Regarding the infection site, higher percentages of XDR strains were observed for those obtained from tissue fragment, followed by urine, tracheal secretion, and blood culture.

We found variations of 45.5% and 100% among the isolates with MDR profiles for the following groups of antimicrobials: second- and third-generation cephalosporins, carbapenems, beta-lactams, aminoglycosides, fluoroquinolone, and glycylcyclines. It was found that 100% of the isolates from the blood, tracheal secretion, and urine exhibited identical resistance profiles at the same classes of drugs, that is, these isolates were resistant to cephalosporines (cefotaxime and cefoxitin), beta-lactams (ampicillin/sulbactam), and glycylcyclines. In addition, one of the urine isolates was resistant to polymyxin E (colistin) (Table 5).

### 3.2. Identification and Detection by Multiplex-PCR of Virulence Genes in Pseudomonas aeruginosa Isolates

The identification of *P. aeruginosa* isolates was performed by PCR assays individually for each gene *oprI* and *oprL*, and all 99 isolates amplified showed fragments of 249 and 504 bp, respectively. In addition, the occurrence of other different virulence genes were screened by mPCR, as illustrated in Figure 1.

With respect to the distribution of the number of virulence genes that were detected in each type of clinical specimen, we observed that all of the virulence genes investigated were detected in isolates recovered from blood culture and in 99.0% of the tracheal secretion samples. The highest detection rates for each gene according to the type of clinical specimen were the *lasB* (89.9%) and *exoY* (87.8%) genes followed by *exoT* (82.8%), *toxA* (81.8%), *algD* (75.7%), *plcN* (75.7%), *phzI* (70.7%), and *plcH* (51.5%). Although less frequent, the presences of genes *exoS* (45.4%), *phzM* (34.3%), *pilB* (27.3%), *apr* (22.2%), *Prot IV* (20.2%), *nan-1* (14.1%), *pilA* (14.1%), and *aprA* (6.0%) were also detected (Table 6).

The statistical analyses showed that the isolates recovered from urine samples had a significant lower proportion of the studied genes when compared to the isolates from other clinical specimens (Kruskal–Wallis test; *p* < 0.05) (Table 6).

### 3.3. Molecular Detection of Genes That Confer Antimicrobial Resistance in Pseudomonas aeruginosa Isolates

The detection of genes that confer resistance to oxacillin (*bla*_OXA-23_*, bla*_OXA-24_*, bla*_OXA-51_*,* and *bla*_OXA-58_), polymyxin E (*mcr-1*), and class C of cephalosporinase (AmpC-β-lactamases) were also investigated in all 99 strains of *P. aeruginosa* (illustrated in Figure 2). A high frequency of *ampC* (97.9%) was observed in this study, and the *bla*_OXA51_ gene was found in three isolates. It is interesting to note that one of these isolates (SLZ183) also presented the specific fragment for the *bla*_OXA23_ gene. The *mcr-1* gene, encoding the colistin resistance protein (polymyxin E), was detected in one of the isolates (24LCM) recovered from a urine sample. This isolate showed resistance to at least one antibiotic of each tested class, except for carbapenems, imipenem, and meropenem, to which it was sensitive.

### 3.4. Sequencing Analysis of bla_OXA-23_, bla_OXA-51_, and mcr-1 Genes and the 16S Ribosomal Region

Due to the novelty of data regarding the occurrence of *bla*_OXA-23,_
*bla*_OXA-51,_ and *mcr-1* in *P. aeruginosa*, the PCR products of the 16S ribosomal region for four isolates positive for these genes were sequenced. The phenograms obtained for each isolate were aligned by the Mega v.6.0 program using sequences of the same genes present in Genbank for the elimination of ambiguities. The consensus sequences were submitted to the Blastn program of the BLAST 2.0 package (Basic Alignment Search Tool—http://www.ncbi.nlm.nih.gov/BLAST (accessed on 23 May 2020)). It obtained nucleotide similarity indexes ranging from 96.38 to 100% for the 16 S ribosomal region and from 99.13 to 100% for the *mcr-1, bla*_OXA-23_, and *bla*_OXA-51_ genes. All these sequences were deposited at Genbank and are available for access. For sequences from the 16 S ribosomal region, the access numbers are MT876550 to MT876553, while for the other genes, the access numbers are MT880235 to MT880239.

### 3.5. Clonal Profile of Pseudomonas aeruginosa Isolates by ERIC-PCR

The electrophoretic analysis of the molecular profiles of the fragments generated by ERIC-PCR and submitted at multivariate analysis by NTSYS showed an average number of three to seven fragments in the range of 100 to 3000 bp. ERIC-PCR analysis allowed for grouping of most of the isolates (66.7%) into thirteen distinct clusters (G01 to G13). In contrast, the other 33 isolates exhibited unique genotypic profiles and they were identified in the dendrogram using square brackets (Table 7 and Figure 3).

## 4. Discussion

Infections caused by *P. aeruginosa* are difficult to treat since this bacterium has several mechanisms of resistance and virulence, such as the acquisition of plasmids or integrons. This results in acquisition genes related to (i) production of enzymes that inactivate or modify the drug targets, (ii) overexpression of efflux pumps, (iii) production of toxic proteins, (iv) biofilm formation, and (v) decreased expression of pore-forming proteins [31]. This panorama denotes the relevance of epidemiological studies employing molecular and phenotypic approaches to characterize the pathogenic profile of the strains in circulation in the hospitals. In this study, the presence of genes related to virulence and antimicrobial resistance was investigated in 99 strains obtained from patients of different hospitals located in São Luís (Maranhão, Brazil).

Overall, the strains exhibited high levels of antimicrobial resistance for the drugs used for the treatment of Gram-negative infections. It was possible to observe one strain with resistance to colistin, while no resistance was detected towards polymyxin B. These results are important since polymyxins are the last therapeutic options for the treatment of multi-resistant Gram-negative bacterial infections [32,33].

In fact, dissemination of the plasmid that confers resistance to colistin (mcr-1) is a recent phenomenon in the literature [34]. Some genetic and structural modifications have been related to the occurrence of colistin resistance in *P. aeruginosa* such as changes in Lipid A; in mutations on *cupC1*, *phoQ*, *oprQ*, *rlpA*, *hpaC*, *lpxC*, and *arnB*, and in other genes related to transport and energy production [34,35].

Several drugs were shown to be ineffective against most of the evaluated strains (piperacillin/tazobactam, ciprofloxacin, ceftazidime, and cefepime), besides those for which *P. aeruginosa* has intrinsic resistance. This resistance may be related to the expression of several drug efflux pumps, the low permeability of its outer membrane and the natural occurrence of chromosomal β-lactamase or *ampC* [3,36].

The multiple resistance profiles observed for these isolates are in concordance with the literature data. For example, several studies have shown high rates of resistance towards the following classes of antimicrobials: cephalosporins (ceftazidime 93% and cefepime 89%) [37]; to carbapenems (Imipenem 41%, meropenem 88%) [38]; aminoglycosides (Amikacin and gentamycin, 91% and 98%, respectively); fluoroquinolone (ciprofloxacin, 98%) [39].

Similarly, the values of detection of *P. aeruginosa* with MDR profiles have varied among studies from different countries: in Malaysia, the value was found to be 19.6% [40]; while studies performed in Nepal [41] and Iran [42] detected rates of 89.4% and 100%, respectively. It is well-known that large-scale use of different classes of antimicrobial agents such as Monobactams, Penicillins, synthetic derivatives of penicillin, aminoglycosides, cephalosporins, fluoroquinolones, carbapenems, and polymyxins in the treatment of bacterial infections has led to the emergence of MDR, XDR, and even pan-drug-resistant (PDR) strains [43,44].

Given the high level of drug resistance in the strains evaluated in this study, the occurrence of genes related to antimicrobial resistance was investigated. It was observed that almost all strains carried the *ampC* gene (97.9%), related to cephalosporins resistance. This alarming incidence of *ampC* (or its overexpression) in clinical isolates of *P. aeruginosa* was also reported by other authors [45,46]. Studies have shown that the rates of molecular detection of *ampC* has varied among clinical isolates of *P. aeruginosa*. These differences between detection rates may be related to mutations affecting the *AmpC* gene or specificity of the different primers or methodologies that have been used for their detection in *P. aeruginosa* [47,48,49]. On the other hand, three isolates carrying the gene *bla*_OXA51_ were detected, and in one of these isolates, the *bla*_OXA23_ gene was also detected. The *mcr-1* gene encoding the colistin resistance protein (polymyxin E) was detected in one of the isolates evaluated in this study.

The presence of the *bla*_OXA23_ and *bla*_OXA51_ genes in *P. aeruginosa* isolates is an unprecedented finding for the literature. These genes encoding Ambler class D oxacillinase (OXA)-like carbapenemase are often detected in *Acinetobacter baumannii* and are related to resistance to oxacilicins, cephalosporins, and carbapenems [50,51,52]. Similarly, since its discovery in *Escherichia coli* [34], the *mcr-1* gene has been described in other microorganisms, including *K. pneumoniae* [53], *Enterobacter aerogenes* [54] and *Enterobacter cloacae* [55]. In Italy, several bacterial species were detected in hospital surfaces with colistin resistance driven by *mcr-1* gene, including in *P. aeruginosa* and *P. putida* [56].

The detection of several genes that confer virulence to *P. aeruginosa* has also been investigated in this study. We observed that the most prevalent genes were *oprI* and *oprL*, which encode outer membrane lipoprotein I (maintenance of cell membrane integrity) and peptidoglycan-associated lipoprotein OprL (maintains the shape of the bacterium), respectively [57]. These results agree with the findings observed for *P. aeruginosa* isolates recovered from burn, pulmonary tract and blood infections [58].

In relation to the high detection rate found for the *lasB* gene, similar results were observed by other researchers [59,60]. Several proteases have been considered an important virulence factor of *P. aeruginosa*. The elastase that is encoded by the *lasB* gene is a metalloprotease that degrades elastin and tissue collagen, breaks down fibrin, and destroys other structural proteins. In this sense, this protein has been implicated in the process of invasion and permanence of that bacterium in tissues [61].

High rates of strains carrying the genes *exoY*, *exoT*, and *exoS* were observed in the strains used in this study. The results obtained for *exoY* (ExoY exotoxin) were higher than those described in recent literature, which report detection of this gene ranging from 14.5% to 72.5% [33,62,63]. Regarding the rate of 82.8% that was observed for *exoT*, this result was higher than those obtained by Fazeli and Momtz (2014) (36.27%) [1] and Mahdavi et al. (2017) (76.32%) [64]. On the other hand, the result of 45.4% found for the *exoS* gene was higher than observed Mahdavi et al. (2017) [64] that found it in 30.9%. However, higher rates of *exoS* were obtained by Fazeli and Momtz (2014) [1] (67.6%) and Yousefi-Avarvand et al. (2015) [62] (65.4%).

Another important virulence factor found in most of the isolates (80.8%) was the *toxA* gene, which encodes the *P. aeruginosa* exotoxin A. This protein is a cytotoxic agent that, similar to diphtheria toxin, inhibits protein biosynthesis, causing extensive damage to tissues and organs [65]. The result found for the *toxA* gene is in agreement with the findings obtained by Yousefi-Avarvand et al. (2015) [62]. However, Fazeli and Momtz (2014) [1] found a rate of 35.29% in *P. aeruginosa* strains recovered from burn infections.

Other virulence genes with high detection rates were *algD* (75.7%), *plcN* (75.7%), and *phzI* (70.7%). The prevalence observed for *algD* was superior to that detected by Benie et al. (2017) [7] in *P. aeruginosa* samples isolated from beef and fish. The product of *algD* (GDP-mannose 6-dehydrogenase) is involved in the synthesis of alginate, a polysaccharide that is the main extracellular homopolymer substance and the main component of *P. aeruginosa* biofilm [11,66]. Alginate synthesis is often associated with chronic lung infections in patients with cystic fibrosis [67]. It is interesting to note that, although the *algD* gene is considered intrinsic to *Pseudomonas aeruginosa*, this gene was detected only in 75.7% of the isolates. However, other studies have also shown variation in the detection rate of the *algD* gene, which probably may be related to the specificity of the primers used and or the origin of the bacterial isolate [68,69,70,71].

The presence of the gene that codifies nonhemolytic (*plcN*) and hemolytic (*plcH*) phospholipases C in the clinical isolates evaluated here was verified in 75.7% and 51.5%, respectively. It is interesting to note that Benie et al. (2017) [7] found a rate of 72.1% for *plcH* in *P. aeruginosa* isolated from animals. On the other hand, in the isolates of the present study, the prevalence of *plcN* was higher than that observed for *plcH*. These two proteins act on the breakdown of phospholipids, and they contribute to bacterial invasion through their cytotoxic effects on neutrophils, playing an important role in the establishment of bacterial infection [1,72].

The *phzI* gene belonging to the phenazine operon was detected in 70.7% of *P. aeruginosa* isolates. This result was lower than the 90% rate found by Ertugrul et al. (2017) [73]. The product encoded by the *phzI* gene acts with other virulence factors (*phzH*, *phzM*, and *phzS*) to destroy host tissue components [74]. The *pilB* gene, which encoded fimbrial-type IV protein, was found in 27 (27.3%) of the isolates. This result was lower than that found in the literature. Relatively varied rates have been found for this gene, such as 35%[75], 73.2% [59] and 75.0% [70], respectively.

The presence of the gene encoding an alkaline metalloprotease (*apr*) was detected in 22.2% of the isolates. This result was superior to that found by Fazeli and Momtz (2014) [74], who detected this gene in 12.7% of its isolates. In contrast, Nahar et al. (2017) [76] in an in silico study analyzing 18 Genbank sequences found the prevalence of *apr* in 66.6%. Similarly, *Prot* IV was detected in 20.2% of the samples. This gene encodes an alkaline protease that acts directly on the cytoplasm of the host cell, being responsible for the destruction of structural proteins and helping to establish infection [33,36].

The results also showed a low prevalence for the *nan-1* gene that codifies a neuraminidase. However, this value was superior to that found by Benie et al. (2017) [7], who did not obtain any positive isolates for this gene. However, Habibi and Honarmand (2015) [70] and Heidary et al. (2016) [59] detected rates of 12.5% and 36.61%, respectively.

The *aprA* gene was detected in 6.0% of the isolates analyzed, which is a rate below the values founds in other studies such as those performed by Mitov, Strateva, and Markova (2010) [68] (91%) and Badamchi et al. (2017) [6] (81.2%). On the other hand, Ertugrul et al. (2017) [73] did not detect the presence of this gene in any tested strains. The product of *aprA* is an elastase that target lymphocytes, neutrophils, and other cells [74].

Analysis of the ERIC-PCR profiles allows for the study of genetic polymorphism among bacterial isolates, determining different ones and grouping similar profiles. In the present study, the phenotypic profile of 99 isolates identified genetically as being of the species *P. aeruginosa* was characterized as heterogeneous, showing 13 different clusters. Thus, a wide genetic diversity among the isolates was observed, although they belonged to the same species, denoting that different *P. aeruginosa* clones circulated in the hospitals from São Luis-MA. Since the ERIC-PCR methodology is fast, sensitive, and reproducible, it has been widely used for the molecular epidemiological analysis of different bacterial species [77,78].

## 5. Conclusions

In the present study, a combination of molecular and phenotypic assays revealed a high level of antimicrobial resistance in 99 *P. aeruginosa* isolates, including a high frequency of strains with MDR and XDR profiles. In addition, the occurrence of bla_OXA-23_, bla_OXA-51,_ and *mcr*-1 genes in *P. aeruginosa* isolates was detected. Furthermore, these strains carry multiple genes related to important virulence pathways in *P. aeruginosa*, including those related to the impermeability of the outer membrane, tissue damage, exotoxin release, and biofilm formation. These strains also exhibited a wide genetic diversity, denoting that different clones are present in the studied area. Taken together, these data reveal the need for constant investigation of the occurrence of strains with multidrug resistance and with high pathogenic potential in order to guide the implantation of effective approaches to reduce their spread and to then improve the quality of health care facilities.

## Figures and Tables

**Figure 1 microorganisms-09-00786-f001:**
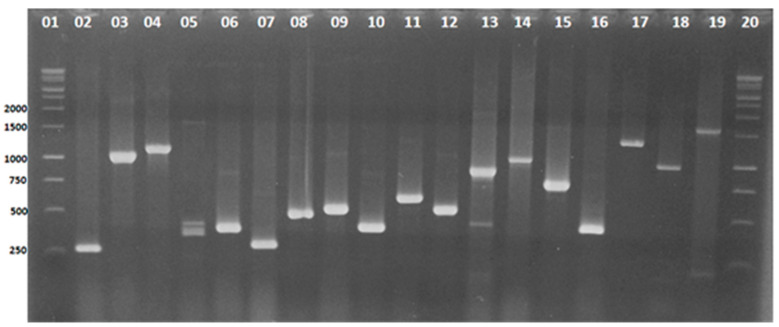
Electrophoretic profiles of the virulence genes investigated in the 99 isolates of *Pseudomonas aeruginosa:* 1.5% agarose gel stained with ethidium bromide. Lines 1 and 20: 1 Kb DNA ladder (Promega, USA). Lines 2 to 19 profiles of the isolates carrying the genes: *oprI* (249 bp), *exoY* (1035 bp), *exoT* (1.159 bp), *lasB* (284 bp), *toxA* (396 bp), *algD* (300 bp), *plcN* (481 bp), *oprL* (504 bp), *phzl* (392 bp), *plcH* (608 bp), *exoS* (504 bp), *phzM* (875 bp), *apr* (1.017 bp), *Prot IV* (752 bp), *pilB* (408 bp), *nan-1* (1.316 bp), *aprA* (993 bp), and *pilA* (1.675 bp).

**Figure 2 microorganisms-09-00786-f002:**
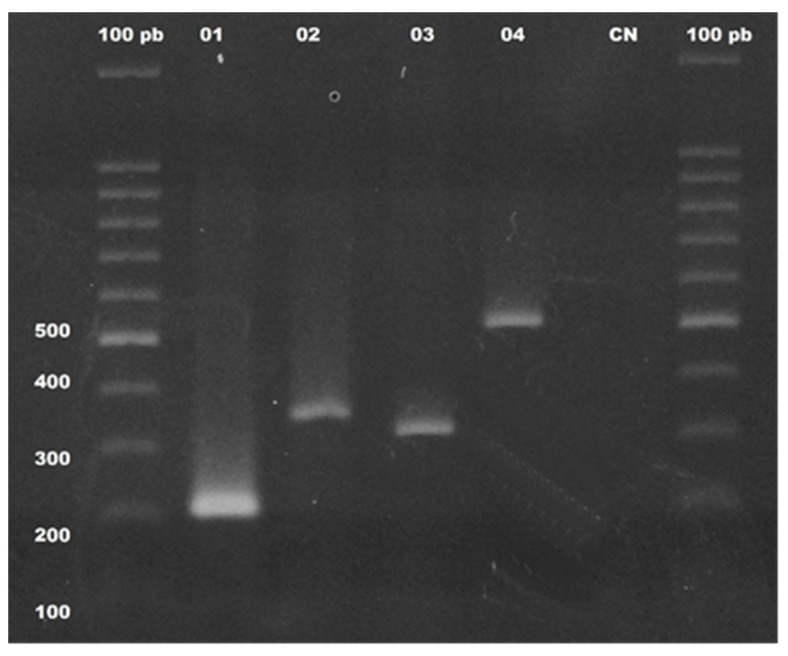
Molecular detection of genes that confer resistance to antimicrobials cephalosporin (*ampC*), oxacillins, and colistin (polymyxin E) in clinical isolates of *Pseudomonas aeruginosa*: 1.5% agarose gel, stained with ethidium bromide, showing the amplification profiles obtained for the *ampC* genes (lines 1–218 bp), *mcr-1* gene (lines 2–400 bp), *bla*_OXA-51_ gene (lines 3–353 bp), and *bla*_OXA-23_ gene (lines 4–501 bp). Molecular marker, 100 bp DNA ladder (Promega, USA) and CN = negative control of the reagents used in the PCR amplification assays.

**Figure 3 microorganisms-09-00786-f003:**
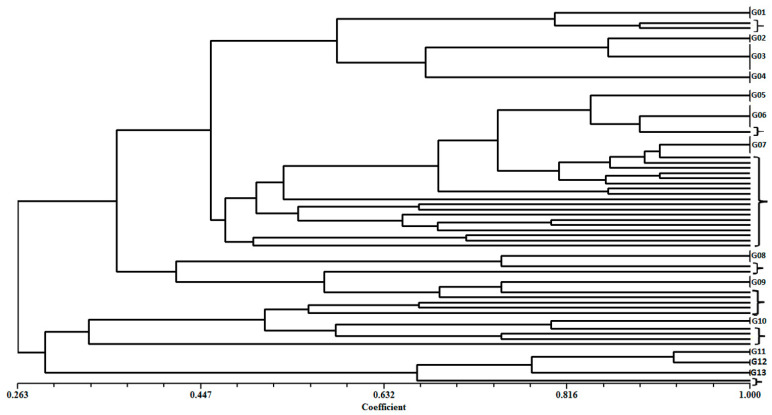
Dendrogram of genetic similarity derived from the DICE similarity coefficient showing the relationships between *Pseudomonas aeruginosa* isolates. Construction made using the NTSYS program with the Unweighted Pair Group Method with Arithmetic Mean (UPGMA) algorithm.

**Table 1 microorganisms-09-00786-t001:** Primers used for the detection of different virulence genes by mPCR in *Pseudomonas aeruginosa* isolates evaluated in this study.

Groups (for mPCR)	Target Gene	Primers Sequence (5′-3′)	Amplicon Size (pb)	Virulence Fator (Gene Product)	References
**Group I**	*exoS*	CTTGAAGGGACTCGACAAGG (F) TTCAGGTCCGCGTAGTGAAT (R)	504	Exotoxin S	[19]
	*exoT*	CAATCATCTCAGCAGAACCC (F) TGTCGTAGAGGATCTCCTG	1159	Exotoxin T	[20]
	*phzI*	CATCAGCTTAGCAATCCC (F) CGGAGAAACTTTTCCCTC (R)	392	Phenazine prodution	[20]
	*phzM*	ATGGAGAGCGGGATCGACAG (F) ATGCGGGTTTCCATCGGCAG (R)	875	Phenazine prodution	[20]
	*apr*	TGTCCAGCAATTCTCTTGC (F) CGTTTTCCACGGTGACC (R)	1017	Protease	[20]
	*pilA*	ACAGCATCCAACTGAGCG (F) TTGACTTCCTCCAGGCTG (R)	1675	Type IV fimbria	[20]
**Group II**	*oprI*	ATGAACAACGTTCTGAAATTCTCTGCT (F) CTTGCGGCTGGCTTTTTCCAG (R)	248	Outer membrane lipoprotein I	[21]
	*oprL*	ATGGAAATGCTGAAATTCGGC (F) CTTCTTCAGCTCGACGCGACG (R)	504	Peptidoglycan-associated lipoprotein	[21]
	*pilB*	TCGAACTGATGATCGTGG (F) CTTTCGGAGTGAACATCG (R)	408	Fimbrial protein	[20]
	*nan*	AGGATGAATACTTATTTTGAT (F) TCACTAAATCCATCTCTGACCCGATA (R)	1316	Putative neuraminidase	[19]
	*algD*	AAGGCGGAAATGCCATCTCC (F) AGGGAAGTTCCGGGCGTTTG (R)	300	GDP-mannose 6-dehydrogenase (alginate production)	[22]
	*plcH*	GCACGTGGTCATCCTGATGC (F) TCCGTAGGCGTCGACGTAC (R)	608	Hemolytic phospholipase C	[23]
	*exoY*	TATCGACGGTCATCGTCAGGT (F) TTGATGCACTCGACCAGCAAG (R)	1035	Adenylate cyclase	[1]
**Group III**	*toxA*	GACAACGCCCTCAGCATCACCAGC (F) CGCTGGCCCATTCGCTCCAGCGCT (R)	396	Exotoxin A	[24]
	*lasB*	GAATGAACGAAGCGTTCTCCGAC (F) TGGCGTCGACGAACACCTCG (R)	284	Protease	[1]
	*aprA*	GTCGACCAGGCGGCGGAGCAGATA (F) GCCGAGGCCGCCGTAGAGGATGTC (R)	993	Alkaline protease	[23]
	*Prot IV*	TATTTCGCCGACTCCCTGTA (F) GAATAGACGCCGCTGAAATC (R)	752	Protease type IV	[25]
	*plcN*	TCCGTTATCGCAACCAGCCCTACG (F) TCGCTGTCGAGCAGGTCGAAC (R)	481	Non-hemolytic phospholipase C	[26]

**Table 2 microorganisms-09-00786-t002:** Primers for the detection of different resistance genes in *Pseudomonas aeruginosa* isolates evaluated in this study.

Groups (for mPCR)	Target Gene	Primer Sequence (5′-3)	Amplicon Size (pb)	Gene Product	Reference
**Group IV**	*bla* _Oxa23_	GATCGGATTGGAGAACCAGA (F) ATTTCTGACCGCATTTCCAT (R)	501	Oxacilinase	[27]
	*bla* _Oxa24_	GGTTAGTTGGCCCCCTTAAA (F) AGTTGAGCGAAAAGGGGATT (R)	246	Oxacilinase	[27]
	*bla* _Oxa51_	TAATGCTTTGATCGGCCTTG (F) TGGATTGCACTTCATCTTGG (R)	353	Oxacilinase	[27]
	*bla* _Oxa58_	AAGTATTGGGGCTTGTGCTG (F) CCCCTCTGCGCTCTACATAC (R)	599	Oxacilinase	[27]
**Group V**	*ampC*	CGGCTCGGTGAGCAAGACCTTC (F) AGTCGCGGATCTGTGCCTGGTC (R)	218	Cephalosporinase	[28]
	*mcr-1*	CAGTATAATTGCCGTAATTATCCCACCGT (F) GTCTCGGCTTGGTCGGTCTGTAG (R)	400	LPS-modifying enzyme	This study

**Table 3 microorganisms-09-00786-t003:** Distribution of *Pseudomonas aeruginosa* isolates according to gender, age group, and number of isolates from the type of clinical specimen.

Gender	Tissue Fragment (*n* = 11)/%		Blood Culture (*n* = 40)/%		Tracheal Secretion (*n* = 37)/%		Urine (*n* = 11)/%		*p* Value
Male	4	36.4	18	45	21	56.8	8	72.7	ns
Female	7	63.6	22	55	16	43.2	3	27.3	ns
Age group (years)									*p* < 0.05
up to 11	0	-	7	17.5	-	-	-	-	
12 to 19	0	-	1	2.5	2	5.4	1	9.1	
20 to 60	3	27.3	9	22.5	11	29.7	3	27.3	
over 60 (years)	8	72.7	23	57.5	24	64.9	7	63.6	

Caption: ns = no significant. *p* value < 0.05: statistical significance.

**Table 4 microorganisms-09-00786-t004:** Antibiotic resistance profile of *Pseudomonas aeruginosa* isolates evaluated in this study as determined by the Vitek 2 Compact system.

Antimicrobial	MIC (mg/mL)	Number of Isolates	% Resistance
Cefoxitin	≥64	99	100.0
Ceftazidime	≥64	70	70.7
Cefepime	≥64	61	61.6
Ampicilin/sulbactam	≥32/16	99	100.0
Piperacilin/tazobactam	≥128/4	74	74.7
Aztreonam	≥32	39	39.4
Ciprofloxacin	≥4	66	66.7
Gentamicin	≥16	58	58.6
Amikacin	≥64	44	44.4
Imipenem	≥16	57	57.6
Meropenem	≥16	56	56.6
Colistin	≥8	1	1.0
Polymyxin B	≤0.5	0	0.0
Tigecycline	≥16	99	100.0

**Table 5 microorganisms-09-00786-t005:** Antimicrobial resistance profile of *Pseudomonas aeruginosa* isolates by infection site.

	Fragment of Tissue		Blood Culture		Tracheal Secretion		Urine		
	No	%	No	%	No	%	No	%	
Amikacin	5	45.5	16	40.0	17	45.9	6	54.5	ns
Gentamycin	6	54.5	17	42.5	25	67.6	10	90.9	<0.05
Cefepime	10	90.9	19	47.5	25	67.6	7	63.6	<0.05
Cefotaxime	11	100.0	40	100	37	100.0	11	100.0	<0.05
Cefoxitin	11	100.0	40	100	37	100.0	11	100.0	<0.05
Ceftazidime	11	100.0	19	47.5	31	83.8	9	81.8	<0.05
Imipenem	10	90.9	9	22.5	30	81.1	8	72.7	<0.05
Meropenem	10	90.9	9	22.5	29	78.4	8	72.7	<0.05
Ampicillin/Sulbactam	11	100.0	40	100	37	100.0	11	100.0	ns
Piperacillin/Tazobactam	9	81.8	25	62.5	30	81.1	10	90.9	ns
Aztreonam	9	81.8	7	17.5	17	45.9	6	54.5	<0.05
Ciprofloxacin	11	100.0	22	55.0	25	67.6	8	72.7	<0.05
Polymyxin B	0	0.0	0	0.0	0	0.0	0	0.0	ns
Polymyxin E	0	0.0	0	0.0	0	0.0	1	1.0	
Tigecycline	11	100.0	40	100.0	37	100.0	11	100.0	ns

Caption: No = number; ns = not significant. *p*-value < 0.05: statistical significance.

**Table 6 microorganisms-09-00786-t006:** Distribution of virulence genes detected in *Pseudomonas aeruginosa* isolates according to the type of clinical specimen.

Clinical Specimen	Genes Detected in Clinical Isolates of *Pseudomonas aeruginosa*																	
	*oprl*	*oprL*	*lasB*	*exoY*	*toxA*	*exoT*	*algD*	*plcN*	*phzl*	*plcH*	*pilA*	*ExoS*	*phzM*	*pilB*	*apr*	*Prot IV*	*nan-1*	*aprA*
Blood culture	40	40	34	32	31	30	23	32	27	11	1	15	13	5	7	5	6	5
Tracheal secretion	37	37	35	35	34	33	32	26	29	25	6	19	13	13	10	10	5	0
Urine	11	11	9	9	6	8	9	8	6	7	2	5	2	3	1	2	3	1
Fragment of tissue	11	11	11	11	10	11	11	9	8	8	5	6	6	6	4	3	0	0
Total genes	99	99	89	87	81	82	75	75	70	51	14	45	34	27	22	20	14	6

**Table 7 microorganisms-09-00786-t007:** Profiles of groups of fragments and number of isolates in each clade formed by multivariate analysis using NTSYS.

Clades	Base Pair Fragments (bp)	Number of Isolates in Each Group
G01	350–500–1000	8
G02	350–650–1000	5
G03	150–350-650–1000	7
G04	150–350-400	13
G05	250–350–750	2
G06	250–350–750–1000	7
G07	250–350–750–1000–2500	5
G08	250–350–750–1000–1300–2500	3
G09	150–250–350–400	5
G10	350–650–850–2000	2
G11	350–400–550–700–800–1100–2000	2
G12	350–400–700–800–1100–2000	2
G13	350–550–800–1100–2000	5
Isolated in brackets	Isolates with heterogeneous profiles with different base pair sizes	33
Total		99

## Data Availability

The data presented in this study are available on request from the corresponding author.

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
