# Peer review of "Molecular Detection of Drug-Resistance Genes of blaOXA-23-blaOXA-51 and mcr-1 in Clinical Isolates of Pseudomonas aeruginosa"

_microorganisms, 2021, doi:10.3390/microorganisms9040786_

Round 1
Reviewer 1 Report
In the manuscript ID microorganisms-1166692 by Nitz and colleagues, the authors describe the phenotypic and molecular characterization of 99 Pseudomonas aeruginosa strains isolated from different clinical specimens. The study evidenced a high rate of multidrug and extensively-drug resistance in the analysed isolates, as well as the carriage of several virulence factors and a wide range of bacterial clones circulating in the hospitals involved in the strains recovery. Of particular interest, the blaOXA-23, blaOXA-51 and mcr-1 genes were identified in the clinical strains, a result reported as first detection for the two bla genes, which, according to literature, have been mainly observed in Acinetobacter baumanii.
The paper is clear, the results are sounding and of great interest for the clinical management of P. aeruginosa infections. The authors have successfully identified a wide variety of genes and the most important ones have been confirmed by sequencing. The results are well supported by statistical analysis and well discussed.
There are just few issues that need to be addressed and some data need corrections:
-Among antibiotic resistance determinants, the authors have sought by PCR assays only those involved in β-lactams and polymyxins resistance, whereas antibiotic susceptibility was assessed phenotypically even towards aminoglycosides and quinolones, reporting relevant percentages of resistant P. aeruginosa strains. Why did the authors choose not to search the resistance determinants against these two antibiotic classes?
The rationale driving their choice should be declared in the discussion section.
-The algD and ampC genes should be intrinsic in P. aeruginosa (Ma et al., 1997 doi: 10.1111/j.1574-6968.1997.tb10291.x; Berrazeg et al., 2015 doi: 10.1128/AAC.00825-15). Is it possible that their failed detection in the analyzed P. aeruginosa strains is due to mutations in the target sequences of the used primer pairs? Please clarify.
-In Table 4, reporting the MIC values on the basis of the Kirby-Bauer test is not correct. Please correct indicating the susceptibility profile (i.e. susceptible, S; intermediate, I; resistant, R).
-In line 221, cefotaxime and cefoxitin are indicated as carabapenems, while they should be defined as cephalosporines.
-“alginate” is not a protein (line 402) but a polysaccharide, please correct.
Once solved these minor revisions, the manuscript can be published in “Microorganisms”.
MINOR COMMENTS
Lines 17, 18, please substitute “minimum 17 inhibitory concentration (MIC)” with “the antibiotic susceptibility profile”;
Lines 25, 273 and 281, please type “mcr-1” in italic;
Line 37, please correct “and causes a variety of systemic infections”;
Line 42, please correct “and increases the competitiveness”;
Lines 55, 56, please correct “(i.e. bacterial isolates remain susceptible to only one or two 55 categories);
Line 76, please delete the term “were” before “collected from November 2015”;
Line 80, please correct “Broth Heart broth” with “Brain Heart broth”;
Line 95, please correct “antimicrobial classes: ß-Lactamics”;
Lines 129, 132, 192, 207 and 224, please type “Pseudomonas aeruginosa” in italic;
Please uniform the table heading titles in Table 1 and 2;
Line 140, please correct “we amplified”;
Line 199, please delete “MIC ≥ 8 mg/mL”;
Line 277, please type “ampC” in italic;
Line 281, please correct “ampC gene”;
Line 291, please correct “Nucleotide similarity indexes were ranging”;
Line 294, please correct “the access numbers for 16S ribosomal regions are”;
Please uniform the font type of Table 7 to the other tables;
Line 312, please correct “This results in the acquisition of genes related to”;
Lines 317, 318, please correct “the presence of genes… was investigated in 99 strains”;
Line 321, please uniform the font type font in “over the age of 60 years”;
Line 329, please delete the term “and” before “polymyxins”;
Line 357, please correct “carried the ampC gene”;
Line 361, please uniform the font type font in “polymyxin E”;
Lines 368-270, please correct “In Italy, several bacterial species were detected in hospital surfaces with colistin resistance driven by the mcr-1 gene, including in P. aeruginosa and P. putida”, and uniform the font type;
Line 373, please delete the terms “a high occurrence”;
Line 400, please type “phzI” in Italic and delete “the” before “algD”;
Line 431, please correct “The aprA gene was detected”;
Line 442, please sustitute the term “for” with “since”;
In the conclusions, please type P. aeruginosa and the gene names in Italic.
Author Response
Dear reviewer,
Thanks a lot for all your contribuitions that enhanced the quality of our manuscript. We have changed the text following your suggestions (please the changes as they are highlighted in yellow).
Following, we provided the response for your questions.
-Among antibiotic resistance determinants, the authors have sought by PCR assays only those involved in β-lactams and polymyxins resistance, whereas antibiotic susceptibility was assessed phenotypically even towards aminoglycosides and quinolones, reporting relevant percentages of resistant P. aeruginosa strains. Why did the authors choose not to search the resistance determinants against these two antibiotic classes?
Our response: The results of phenotypic analysis for susceptibility are already standardized in our laboratories, because we follow the standards for the release of reports. In addition, we also have previous data that show the dissemination of strains resistant to carbapenems and emergent for resistance to polymyxins in hospitals in our region.
-The algD and ampC genes should be intrinsic in P. aeruginosa (Ma et al., 1997 doi: 10.1111/j.1574-6968.1997.tb10291.x; Berrazeg et al., 2015 doi: 10.1128/AAC.00825-15). Is it possible that their failed detection in the analyzed P. aeruginosa strains is due to mutations in the target sequences of the used primer pairs? Please clarify.
Our response: Some other studies have also not detected the gene algD in all isolates (Valadbeigi et al. DOI: 10.2174/1871526517666170427121956; Elmouaden et al. DOI: 10.3855/jidc.10675). The literature also reports a variety of mutations on ampC that could explain the absence of detection of this genes.
-In Table 4, reporting the MIC values on the basis of the Kirby-Bauer test is not correct. Please correct indicating the susceptibility profile (i.e. susceptible, S; intermediate, I; resistant, R).
Our response: We have evaluted the antimicrobial susceptibility by two methods (automated Vitek®2 Compact system and disc diffusion method). The results expressed in the table 4 is from automated Vitek®2 Compact system.
Reviewer 2 Report
Dear authors,
the revised article reports on a interesting study of antibiotic resistance of P.aeruginosa strains. ALthough it is well know the resistance phenomenon to many classes of drugs, I think that is is always useful to add information of strain resistance variability on susceptibility. Moreover, the genetic studies can in a such a way supports all these data. In my opinion the results are interesting and ready for publication. However, I suggest to specify better when the PCR have been done in simplex or multiplex. the results are well supported by literature in the discussion. Only minor revision are requested. I attach the requested revisions in the attached article (in pdf).

Author Response
Dear reviewer,
Thanks a lot for all your contribuitions that enhanced the quality of our manuscript. We have changed the text following your suggestions (please the changes as they are highlighted in yellow).